# Return Capping: Sample-Efficient CVaR Policy Gradient Optimisation

**Harry Mead** [1]   **Clarissa Costen** [1]   **Bruno Lacerda** [1]   **Nick Hawes** [1]

## Abstract

When optimising for conditional value at risk (CVaR) using policy gradients (PG), current methods rely on discarding a large proportion of trajectories, resulting in poor sample efficiency. We propose a reformulation of the CVaR optimisation problem by capping the total return of trajectories used in training, rather than simply discarding them, and show that this is equivalent to the original problem if the cap is set appropriately. We show, with empirical results in an number of environments, that this reformulation of the problem results in consistently improved performance compared to baselines. We have made all our code available here: https://github.com/HarryMJMead/cvar-return-capping.

## 1. Introduction

In applications that are safety critical, or where catastrophic failure is a possibility, it can be beneficial to minimise the worst case outcomes of decisions, rather than simply performing well on average. Risk-averse reinforcement learning (RL) addresses this problem by maximising the performance of a policy using a *risk metric* as the objective, rather than expected value. We focus here on *conditional value at risk* (CVaR). CVaR represents an intuitive, coherent (Artzner et al., 1999) risk metric that assesses the expected value of the worst $\alpha$ proportion of runs. Note that in this work we focus on *static* CVaR, where static CVaR refers to the CVaR of full episode returns, due its ease of interpretability for setting a level of risk aversion.

A common approach to maximising CVaR in RL is to use policy gradient (PG) methods (Tamar et al., 2015). In the case of CVaR, PG methods sample a set of trajectories, then discard all but the bottom $\alpha$-proportion based on total returns, and then aim to maximise the expected return of this remaining subset of trajectories. Whilst this is sufficient for

some applications, it presents a number of issues. Primarily, it results in *very poor sample efficiency*, especially when $\alpha$ is low, as a large proportion of sampled trajectories are discarded. Figure 1(a) illustrates how existing CVaR PG methods are inefficient in terms of trajectory usage. In addition, since it is the highest return samples that are discarded, *the policy is unable to learn from the best performing trajectories*, which can present further issues for policy training. In order to mitigate these issues, we propose *Return Capping*. Rather than discarding trajectories, we propose that all trajectories are used in training, but the return of those above a certain threshold is truncated. We prove in Section 4 that if this threshold, or cap, is set correctly, optimising CVaR under Return Capping is equivalent to optimising CVaR without the cap. Since Return Capping allows all trajectories to be used, sample efficiency is greatly improved, and learning performance improvements from high-performing trajectories is possible. Our main contributions are:

- Proposing the Return Capping reformulation of the CVaR optimisation objective, with proof that the two are equivalent if the cap is set correctly.

- Presenting a method for approximating this target cap.

- Performing empirical evaluations of this method in established risk-aware benchmarks, where Return Capping has better performance than state-of-art baselines.

We also introduce a new CVaR policy gradient baseline based on PPO, which we show performs better than standard baselines in some environments.

## 2. Related Work

Training for risk-averse behaviour has been explored in many existing works. Direct reward shaping has been used to explicitly discourage risky behaviours (Wu et al., 2023). Work in robust adversarial RL (Pinto et al., 2017; Pan et al., 2019) involves simultaneously training both the primary agent and an adversary that perturbs environment dynamics in order to train more robust policies. However, in both of these approaches, it is challenging to quantify the level of risk-aversion that results from either the adversary or reward shaping.

[1]University of Oxford. Correspondence to: Harry Mead <hmead@robots.ox.ac.uk>.

*Proceedings of the $42^{nd}$ International Conference on Machine Learning*, Vancouver, Canada. PMLR 267, 2025. Copyright 2025 by the author(s).

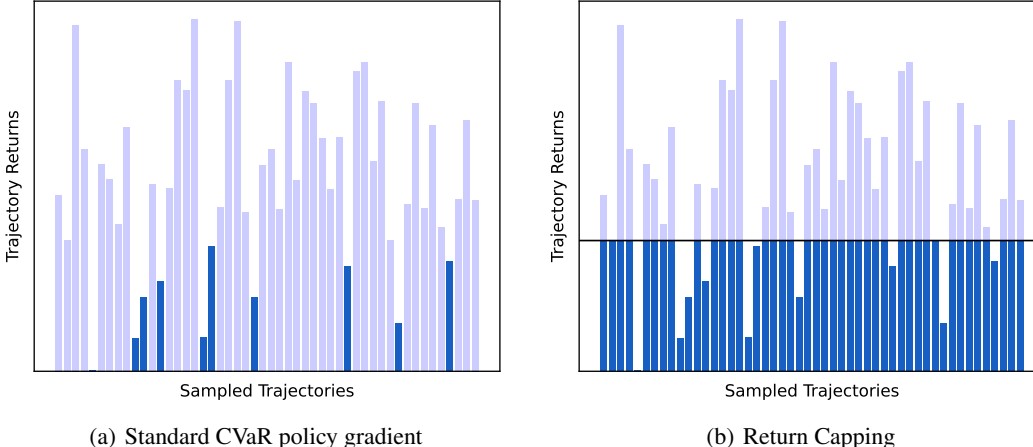

(a) Standard CVaR policy gradient

(b) Return Capping

*Figure 1.* Comparison of trajectory usage of an example batch of sampled trajectories between standard CVaR Policy Gradient methods and Return Capping

A method of quantifying risk aversion is to use risk metrics. Possible metrics include mean variance (Sobel, 1982; Sato et al., 2001; La & Ghavamzadeh, 2013; Prashanth & Ghavamzadeh, 2016) or Value at Risk (VaR) (Filar et al., 1995; Chow et al., 2018), however these metrics are not *coherent* (Artzner et al., 1999), which can present issues when using them for decision making. Examples of coherent risk metrics include entropic value at risk (Ahmadi-Javid, 2012), the Wang risk measure (Wang, 2000), or, the focus of this work, conditional value at risk (CVaR) (Rockafellar et al., 2000). Optimising CVaR using policy gradients has been explored in a number of works (Tamar et al., 2015; Tang et al., 2019; Keramati et al., 2020; Schneider et al., 2024; Kim & Min, 2024). However, much of this work is based Distributional-RL (DRL) (Bellemare et al., 2017), and generally focuses on dynamic CVaR (Dabney et al., 2018; Ma et al., 2020; Lim & Malik, 2022; Schneider et al., 2024). Unlike static CVaR, dynamic CVaR is Markovian (Ruszczyński, 2010), and thus can be locally optimised in a policy. However, it is difficult to interpret and can lead to overly conservative or overly optimistic behaviours (Lim & Malik, 2022). (Lim & Malik, 2022) does suggest a modification to DRL to optimise for static CVaR, but results in (Luo et al., 2024) show poor performance in relation to policy gradient methods.

The most similar works to ours, with a focus on efficient optimisation of static CVaR are (Greenberg et al., 2022) and (Luo et al., 2024). (Greenberg et al., 2022) shows significant sample-efficiency improvements over baselines but requires the environment to be formulated as a Context-MDP (Hallak et al., 2015), where the context captures part, or all, of the environment's randomness. To account for vanishing gradients, (Greenberg et al., 2022) proposes Soft Risk, where the policy is initially optimised for a risk-neutral

objective, before annealing the risk-sensitivity to the target value. Alternatively, (Luo et al., 2024) presents a policy-mixing approach, where the final policy is a mixture of a risk-neutral and risk-averse policy. However, the benefits of this method are greatest only in environments where there are large regions where the optimal risk-neutral and risk-averse policy are the same, which cannot generally be assumed to be the case.

An alternate approach for risk-averse RL is Constrained RL. Constrained RL presents a problem formulation for settings where distinct reward and cost functions exist, where the objective is to maximise reward subject to constraints on the cost (Stooke et al., 2020; Yang et al., 2021; Wu et al., 2024). However, explicitly separate reward and cost functions limits the applicability of these methods to certain environments, and our work instead focuses on risk over a single optimisation metric.

## 3. Background

### 3.1. Conditional Value at Risk (CVaR)

Let $Z$ be a bounded random variable with a cumulative distribution function $F(z) = P(Z \leq z)$. In the case of this paper, the random variable $Z$ will be the return of a trajectory obtained from an agent acting in an environment. For $Z$, the *value at risk* (VaR) at confidence level $\alpha$ is defined as

$$\text{VaR}_\alpha(Z) = \min\{z | F(z) \geq \alpha\}. \tag{1}$$

The *conditional value at risk* (CVaR) at confidence level $\alpha$ is then defined as

$$\text{CVaR}_\alpha(Z) = \frac{1}{\alpha} \int_0^\alpha \text{VaR}_x(Z) \, dx. \tag{2}$$

Alternatively, if $Z$ is continuous, the CVaR can be expressed as

$$\text{CVaR}_\alpha(Z) = \mathbb{E}[Z|Z \leq \text{VaR}_\alpha(Z)]. \tag{3}$$

Therefore, the $\text{CVaR}_\alpha(Z)$ can be interpreted as the expected value of the variable $Z$ in the bottom $\alpha$ proportion of the tail. In our case, the $\text{CVaR}_\alpha(R)$ is the expected total return $R$ of the worst performing $\alpha$ proportion of trajectories.

### 3.2. CVaR Reinforcement Learning

Consider a Markov Decision Process (MDP) defined by $(S, A, P, R)$, corresponding to the state space, the action space, the state transition probabilities and the state transition rewards respectively. Optimal static CVaR policies may be non-Markovian, as static CVaR is dependant on the trajectory history. To account for this, we instead solve for an augmented MDP with a state space $S^+ = (S, \sum r)$, where $\sum r$ is the sum of all rewards up to the current time-step in the trajectory (Lim & Malik, 2022). From this MDP, a policy $\pi_\theta$, parametrised by $\theta$, maps a state input to a distribution over possible actions. With this policy, we can sample trajectories $\tau = \{(s_i, a_i, r_i)\}_{t=0}^T$, and compute the total discounted return $R(\tau) = \sum_{t=0}^T \gamma^t r_t$ using a discount factor $\gamma$. We denote the expected return of a policy as

$$J(\pi_\theta) = \mathbb{E}_{\tau \sim \pi_\theta}[R(\tau)]. \tag{4}$$

Similarly, we denote the policy CVaR at confidence level $\alpha$ as

$$J_\alpha(\pi_\theta) = \mathbb{E}_{\tau \sim \pi_\theta}[R(\tau)|R(\tau) \leq \text{VaR}_\alpha(R(\tau)]. \tag{5}$$

Given a set of trajectories $\{\tau_i\}_{i=0}^N$, the CVaR policy gradient (CVaR-PG) method aims to maximise $J_\alpha(\pi_\theta)$ by performing gradient ascent with respect to the policy parameters $\theta$. The CVaR gradient estimation is given by (Tamar et al., 2015)

$$\nabla_\theta \hat{J}_\alpha \left( \{\tau_i\}_{n=0}^N ; \pi_\theta \right) =$$
$$\frac{1}{\alpha N} \sum_{i=0}^N \Bigg( \mathbf{1}_{R(\tau_i) \leq \text{VaR}_\alpha}(R(\tau_i)$$
$$- \text{VaR}_\alpha) \sum_{t=0}^T \nabla_\theta \log \pi_\theta(a_{i,t}; s_{i,t}) \Bigg). \tag{6}$$

In this formulation, the indicator function means that only samples from the bottom $\alpha$-proportion of runs are used to compute the gradient.

### 3.3. Limitations of CVaR-PG

The primary issue with the standard method for CVaR-PG optimisation is the poor sample efficiency, especially in cases where the target $\alpha$ is small. For an example of where

---

**Algorithm 1** Return Capping

> **Input:** risk level $\alpha$, batch size $N$, training steps $M$, minimum cap $C^M$, cap step size $\eta$
> **Initialise:** policy $\pi_{\theta_1}$, value function $V_{\theta_2}$, $C \leftarrow C^M$
> **for** $m \in 1 : M$ **do**
>     // Sample Trajectories
>     $\{\tau_i\}_{i=1}^N \leftarrow \text{sample\_trajectories}(\pi_{\theta_1})$
>     // Cap Trajectories
>     $\{\tau_i^c\}_{i=1}^N \leftarrow \text{cap\_trajectories}(C, \{\tau_i\}_{i=1}^N)$
>     Update $\theta_1, \theta_2$ using $\text{PPO}\left( \pi_{\theta_1}, V_{\theta_2}, \{\tau_i^c\}_{i=1}^N \right)$
>     $\text{VaR} \leftarrow \text{calculate\_VaR}(\alpha, \{\tau_i\}_{i=1}^N)$
>     $C \leftarrow C + \eta(\text{VaR} - C)$
>     $C \leftarrow \max(C, C^M)$
> **end for**

---

$\alpha = 0.05$, 95% of all trajectories will be discarded, and so a much greater number samples are required to get an equivalent batch size to when optimising for expected value. Another potential issue with standard CVaR-PG optimisation is *blindness to success* (Greenberg et al., 2022). Given a batch of sampled trajectories, the standard CVaR policy gradient does not differentiate between high trajectory returns due to effective actions being sampled or environment stochasticity benefitting the agent. However it is likely, especially early in training, that a proportion of these high returns are due to better policy actions sampled in those specific trajectories. Due to the nature of CVaR-PG, if these trajectories fall outside of the bottom-$\alpha$ proportion of runs, the policy training is effectively blind to these successes. A related issue is a vanishing gradient when the tail of the distribution of trajectory returns is sufficiently flat. In environments with discrete rewards, it is possible that the bottom $\alpha$ proportion of rewards are all equally bad, resulting in the CVaR-PG gradient vanishing entirely. Similarly to the blindness to success, gradient vanishing is most likely to be an issue early in training. The solution presented in (Greenberg et al., 2022) for both issues is to initially optimise for a risk-neutral policy and then slowly anneal $\alpha$ to the target value, but this can present issues when the risk-neutral and the CVaR optimal policies are sufficiently distinct as policy may become stuck in the local optimal of the risk-neutral policy.

## 4. Return Capping

Rather than discard all trajectories outside of the bottom proportion, instead we propose that we keep all trajectories but truncate all returns above a cap $C$. Algorithm 1 outlines the implementation of this method, and Figure 1 illustrates the difference in how the trajectories are handled compared to standard CVaR-PG. Below, we show that, if this cap $C$ is set correctly, the optimal policy for this problem is

equivalent to the optimal $\text{CVaR}_\alpha$ policy.

**Proposition 4.1.** *Suppose $\pi^*$ is the optimal $CVaR_\alpha$ policy, and $VaR_\alpha(\pi^*)$ is the VaR of this policy. Any policy that satisfies $\pi = \arg\max_\pi \mathbb{E}_{\tau\sim\pi}[\min(R(\tau),C)]$ where $C = VaR_\alpha(\pi^*)$ will also be $CVaR_\alpha$ optimal.*

*Proof.* Below is an expression for the expectation of the capped trajectories.

$$J^C(\pi_\theta; C) = \mathbb{E}_{\tau\sim\pi_\theta}[\min(R(\tau), C)]. \quad (7)$$

If we denote $\text{VaR}_x(\pi)$ as the VaR of confidence level $x$ of the returns of trajectories sampled using the policy $\pi$, we can rewrite this equation as

$$J^C(\pi_\theta; C) = \int_0^1 \min(\text{VaR}_x(\pi_\theta), C)\, dx. \quad (8)$$

Suppose now we set $C = \text{VaR}_\alpha(\pi^*)$, and split the integral into two regions separated by $\alpha$

$$J^C(\pi_\theta; \text{VaR}_\alpha(\pi^*))$$
$$= \int_0^\alpha \min(\text{VaR}_x(\pi_\theta), \text{VaR}_\alpha(\pi^*))\, dx$$
$$+ \int_\alpha^1 \min(\text{VaR}_x(\pi_\theta), \text{VaR}_\alpha(\pi^*))\, dx. \quad (9)$$

Consider the first half of the equation.

$$\int_0^\alpha \min(\text{VaR}_x(\pi_\theta), \text{VaR}_\alpha(\pi^*))\, dx$$
$$\leq \int_0^\alpha \text{VaR}_x(\pi_\theta)\, dx = \alpha J_a(\pi_\theta). \quad (10)$$

Therefore, the upper bound of the first half of the integral is

$$\max_\pi \alpha J_\alpha(\pi) = \alpha J_\alpha(\pi^*). \quad (11)$$

As $\text{VaR}_x(\pi)$ is monotonically increasing with respect to $x$, when $x \leq \alpha$, $\text{VaR}_x(\pi^*) \leq \text{VaR}_\alpha(\pi^*)$, and in the left integral $x \leq \alpha$ for all $x$. Therefore, the optimal $\text{CVaR}_\alpha$ policy $\pi^*$ results in

$$\int_0^\alpha \min(\text{VaR}_x(\pi^*), \text{VaR}_\alpha(\pi^*))\, dx$$
$$= \int_0^\alpha \text{VaR}_x(\pi^*)\, dx = \alpha J_a(\pi^*) \quad (12)$$

and so the left integral is equal to its upper bound. Now considering the right integral

$$\int_\alpha^1 \min(\text{VaR}_x(\pi_\theta), \text{VaR}_\alpha(\pi^*))\, dx$$
$$\leq \int_\alpha^1 \text{VaR}_\alpha(\pi^*)\, dx = (1-\alpha)\text{VaR}_\alpha(\pi^*) \quad (13)$$

and so now we have $(1-\alpha)\text{VaR}_\alpha(\pi^*)$ as an upper bound on the right integral. For the right integral, $x \geq \alpha$ for all $x$, so $\text{VaR}_x(\pi^*) \geq \text{VaR}_\alpha(\pi^*)$ for all $x$. Therefore, the optimal $\text{CVaR}_\alpha$ policy $\pi^*$ results in

$$\int_\alpha^1 \min(\text{VaR}_x(\pi^*), \text{VaR}_\alpha(\pi^*))\, dx$$
$$= \int_\alpha^1 \text{VaR}_\alpha(\pi^*)\, dx = (1-\alpha)\text{VaR}_\alpha(\pi^*) \quad (14)$$

and so is also equal the right integral upper bound. Therefore,

$$\max J^C(\pi; \text{VaR}_\alpha(\pi^*)) = J^C(\pi^*; \text{VaR}_\alpha(\pi^*)) \quad (15)$$

and so the optimal $\text{CVaR}_\alpha$ policy is also the optimal policy to maximise the expected return of the truncated trajectories. As this policy satisfies the upper bound defined by Equations 11, and 13, all optimal policies must satisfy this upper bound. Therefore, from Equation 11, the optimal policy to maximise the expected return on the truncated trajectories must also be the optimal $\text{CVaR}_\alpha$ policy, when the cap is set to $\text{VaR}_\alpha(\pi^*)$. $\square$

### 4.1. Cap Approximation

Whilst we have shown equivalence between the two problems when the cap is set to $\text{VaR}_\alpha(\pi^*)$, in most environments, this value will be unknown and must be approximated. In order to approximate this value, we use the set of trajectories we have previously sampled.

For the trajectories $\left[\{\tau_i\}_{i=0}^N\right]_k$ sampled using the policy $\pi_{\theta_k}$, where $k$ denotes the number of policy gradient updates, a possible approximation is $C = \text{VaR}_\alpha(\pi_{\theta_{k-1}})$. However, this is likely to be a high-variance approximation of $\text{VaR}_\alpha(\pi^*)$, especially with smaller batch sizes. In order to reduce variance, we used the following update rule to update the approximation after each policy update step.

$$C_k = C_{k-1} + \eta\left(\text{VaR}_\alpha(\pi_{\theta_{k-1}}) - C_{k-1}\right) \quad (16)$$

where $\eta$ parametrises the size of the cap update step.

In order to reduce the effect of vanishing gradients, we also introduced a minimum cap value $C^M$. In many environments, it is relatively simple to determine the $\text{CVaR}_\alpha$ of an overly conservative policy that does nothing, e.g. betting 0 at each turn in the betting game outlined below in Section 5.3. We suggest that any environment where finding a risk-averse policy is compelling should have a policy that performs better than this highly conservative baseline. By setting the minimum cap value above the performance of this baseline, we reduce the likelihood of vanishing gradients. However, it is necessary to ensure that $C^M$ is a

valid minimum, where we define a valid minimum cap as $C^M \leq \text{VaR}_\alpha(\pi^*)$. Given,

$$\text{CVaR}_\alpha(\pi) \leq \text{VaR}_\alpha(\pi), \quad \forall \pi \qquad (17)$$

and

$$\text{CVaR}_\alpha(\pi) \leq \text{CVaR}_\alpha(\pi^*), \quad \forall \pi, \qquad (18)$$

the $\text{CVaR}_\alpha$ of *any* policy is a valid minimum cap. Therefore, we know that the $\text{CVaR}_\alpha$ of the conservative baseline is valid. In order to reduce the likelihood of vanishing gradients, we assume a cap marginally higher is also valid, but further increasing of the minimum increases the probability of the cap being invalid. Alternative minimum caps that would guarantee validity include the $\text{CVaR}_\alpha$ of a random policy, or the $\text{CVaR}_\alpha$ of the optimal risk-neutral policy, although the latter may be non-trivial to determine.

### 4.2. Reward Adjustment

Given the objective function in Equation 7, the environment rewards needs to be adjusted so as sum to the capped reward. Given reward $r_t$ at timestep $t$, let

$$R_t = \sum_{i=0}^{t} r_i \qquad (19)$$

To calculate the adjusted reward $r_t^a$

$$r_t^a = \min(R_t, C) - \min(R_{t-1}, C) \qquad (20)$$

Using this adjusted reward, we can use reward-to-go to compute the policy gradients, rather that just using total episode return.

## 5. Experiments

### 5.1. Baselines

Although (Tamar et al., 2015) derives Equation 6 as the policy gradient for CVaR optimisation, empirical evidence suggests this baseline performs poorly in practice (Greenberg et al., 2022; Luo et al., 2024). This outcome is not unexpected, given it both does not incorporate a state-value function baseline (Sutton & Barto, 1998), and also bases the gradient update on total episode return, rather than return-to-go, making credit assignment more difficult. In order to present more representative baselines, we choose to compare return capping to the following:

- *Expected Value*: PPO (Schulman et al., 2017) maximising the expected value of return;

- *CVaR-PG*: Unmodified CVaR-PG from Equation 6; and

- *CVaR-PPO*: A CVaR-PG implementation using PPO (Outlined in Appendix A).

When evaluating Return Capping against these baselines, our focus is on CVaR in relation to total number of agent steps in the environment. As the CVaR-PG methods involve discarding a proportion of trajectories, either each gradient update step will have a smaller batch size compared to Return Capping, or the total number of gradient update steps will be reduced. In order to fairly assess the performance of the CVaR-PG baselines, we will compare Return Capping the better performing of these two options. We have included details on all hyperparameters used for baselines and Return Capping in the Appendix.

### 5.2. Environments

We use a range of environments to compare the reward-capping algorithm to the baseline methods. In selecting illustrative environments, we focus on environments where the CVaR optimal policy is significantly different from the optimal expected value policy. We have outlined these environments below, with further detail in the Appendix.

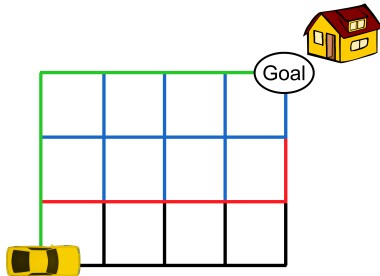

*Figure 2.* Autonomous Vehicle navigation environment from (Rigter et al., 2021). Colours indicate the road type as follows - black: *highway*, red: *main road*, blue: *street*, green: *lane*

### 5.3. Betting Game

The betting game environment (Bäuerle & Ott, 2011) represents a useful domain to assess agents aiming to optimise CVaR due to the substantial and obvious difference between how the agent should act depending on aiming to optimise either expected value or CVaR. In the environment, the agent begins with 16 tokens and is able to wager a proportion of these tokens with $p(win) = x$. If the agent wins, it will win the number of tokens wagered, but will lose the same amount on a loss. The game is sequential, such that after each bet, the subsequent bet is based on the current total of tokens the agent has. The game continues for a fixed number of steps; in our experimentation, six. For $x > 0.5$, the optimal policy to maximise expected is to always bet all

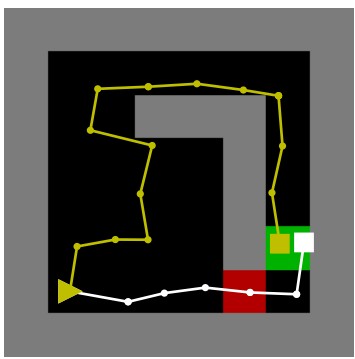

*Figure 3.* Continuous Guarded Maze Environment. Two trajectories shown here - one taking the shortest path passing through the guard whereas the other takes a longer path that avoids the guard

available tokens at each step. Such a policy results in high return trajectories infrequently the majority of trajectories will result negative return. A CVaR policy should aim for even the worst performing runs to result in positive return. For our implementation, $x = 0.8$.

### 5.4. Autonomous Vehicle

The second environment used is adapted from (Rigter et al., 2021). An autonomous car must navigate along different types of roads to reach the goal state, represented as a graph shown in Figure 2. The cost of traversing a given road is sampled from a distribution that depends on the type of road it is, being either *highway*, *main road*, *street* or *lane*. The exact distributions sampled from are outlined in Appendix C.

### 5.5. Guarded Maze

Pictured in Figure 3, the agent in the Guarded Maze aims to reach the green goal state, receiving a reward for doing so. However, should the agent pass through the red guarded zone, it will receive a cost penalty sampled from a random distribution. We adapt two versions of the Guarded Maze from (Greenberg et al., 2022) and (Luo et al., 2023; 2024), which vary in respect to state space and the guard cost distribution. The guard cost distribution is parameterised such that an optimal risk-neutral policy would ignore the guard and always take the shortest path to the goal, whereas a CVaR optimal policy instead should take the longer path to avoid the guard. We have modified both these environments in order to make them more challenging, with these modifications outlined below.

The Guarded Maze, shown in Figure 3, from (Greenberg et al., 2022) consists of a *continuous* state space but discrete action space. The agent is penalised for each step it takes

in the environment, however in (Greenberg et al., 2022), this penalty is limited to only the first 32 environment steps. This results in the CVaR of a policy that does not reach the goal but also does not ever cross the guard being higher than the CVaR of a policy that optimally takes the shortest path passing through the guard. This reduces the likelihood that a policy optimising CVaR would converge to this shortest path. This is because with this limit on step penalties it is no longer optimal compared to random exploration. In our experimentation, we remove this limit and demonstrate that Return Capping still converges to the CVaR-optimal policy.

A *discrete* state space variant of the Guarded Maze was introduced by (Luo et al., 2023). However, the maze used in (Luo et al., 2023) moved the goal such that the difference between the shortest path (9 steps) and the optimal path avoiding the guard (11 steps) was much smaller. This reduces the challenge of policy optimisation since it reduces the likelihood of policies converging to the local optimal of the risk-neutral policy. In our implementation, the goal is positioned as shown in the continuous example shown in Figure 3, resulting in a much greater difference between the shortest (6 steps) and guard avoiding (14 steps) paths.

### 5.6. Lunar Lander

We also evaluate on Lunar Lander from OpenAI Gym (Brockman et al., 2016; Towers et al., 2024). Whilst the other benchmarks have relatively simple environment dynamics, Lunar Lander represents a more complex physics-based environment where the optimal risk-neutral policy is non-trivial. In order to induce a difference between optimal risk neutral and risk-averse policies, we follow the example of (Luo et al., 2023) where the environment is modified so that landing on the right-hand side results in an additional high-variance reward. (Luo et al., 2023) uses a zero-mean reward sampled from $\mathcal{N}(0, 1) * 100$. However, a reward such as this does not necessitate a difference between the risk-neutral and CVaR optimal policies, as a policy that landed on the left-hand side of the landing pad could be optimal both for CVaR and expected value of return. To fix this, we modify the environment such that in addition to the zero-mean, high-variance random reward, landing on the right-hand side also results in an additional 70 reward.

## 6. Results

Figures 4, 5, and 6, show the performance of Return Capping against the baselines outlined above. We also plot the performance of Return Capping depending on the initialisation of $C^M$, using either the VaR of the optimal CVaR policy, the CVaR of the optimal expected value policy, and either slightly above the CVaR of the conservative 'do nothing' policy or the CVaR of a random policy when the former does not exist.

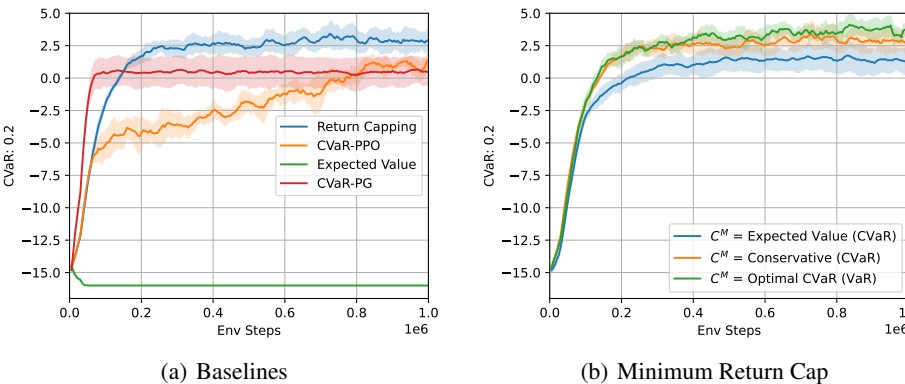

(a) Baselines

(b) Minimum Return Cap

*Figure 4.* Betting Game $\text{CVaR}_{\alpha=0.2}$ performance. Comparing to Baselines and showing effect of different minimum return caps

**Betting Game:** In the betting game, the optimisation objective was $\text{CVaR}_{\alpha=0.2}$. For comparing to the baselines, we initialised $C^M$ based on the CVaR of the conservative policy of betting nothing at each turn. We see in Figure 4(a) that neither CVaR-PPO or CVaR-PG match the performance of Return Capping. As the probability of winning was sufficiently high, the optimal CVaR policy is not trivially to bet nothing each turn. However, we see that when using CVaR-PG, training converges to this overly conservative policy, resulting in a CVaR of close to zero. From Figure 4(b), we see that performance was affected by $C^M$ initialisation. However, all three initialisations result in better performance than the baselines.

**Autonomous Vehicle:** In the autonomous vehicle domain, the optimisation objective was $\text{CVaR}_{\alpha=0.05}$. $C^M$ was initialised using a random policy. In this domain, CVaR-PPO performs very similarly to Return Capping, whereas CVaR-PG takes significantly longer to converge. However, of the eight seeds we ran in this domain, two of the CVaR-PPO runs, did not converge to goal-reaching policies. We have chosen to exclude these two seeds from the plot in Figure C to present a more representative baseline (see Appendix C for the plot without outliers removed). However, the presence of these outliers suggests the Return Capping may be a more robust method compared to CVaR-PPO. Figure 5(d) shows that that the $C^M$ initialisation had minimal effect on the training performance.

**Guarded Maze:** In the continuous state space setting, Figure 5(b) shows that Return Capping outperforms all baselines. For each method, we ran six seeds. None of CVaR-PG runs converged to a policy that reached the goal, and of the CVaR-PPO seeds, two converged to the CVaR-optimal policy, one converged to the risk-neutral policy, and three did not converge to goal-reaching policies. All six of the Return Capping runs converged to the CVaR-optimal policy, again indicating greater training robustness compared to other CVaR-PG methods. In the discrete state space set-

ting, we see in Figure 5(c) that although CVaR-PPO does consistently converge to the optimal policy, Return Capping converges in approximately half as many environment steps. Interestingly, Figures 5(e) and 5(f) show the in both guarded maze domains, the final policy is highly sensitive to $C^M$. Unlike in the two previous domains, where setting $C^M = \text{VaR}_\alpha(\pi^*)$ produced the best results, here we see the opposite. This behaviour is likely due to the difference in complexity of the risk-neutral and risk-averse policies. As mentioned in Section 5.5, the optimal risk-averse path is approximately twice as long as the risk-neutral path. When the cap is high, we see this results in the policy converging to the local, less complex risk neutral policy, reducing policy exploration, and resulting in the risk-averse policy never being discovered. However, when the cap is set sufficiently low, the policy is initially agnostic to trajectory length, given the trajectory return is above the cap. This allows for sufficient exploration for the optimal risk-averse policy to be found.

**Lunar Lander:** In the lunar lander environment, unlike the other environments, initialising the cap minimum using the conservative policy did not result in consistent convergence to the optimal CVaR policy. Figure 6(b) illustrates the relative performance of the three cap minimums. For comparisons to the baselines, we have used $C^m$ equal to the CVaR of the risk-neutral policy. However, as this policy is non-trivial to determine, we have also included an offset Return Capping plot, accounting for the $4e5$ environment steps required to learn the risk-neutral policy. Figure 6(a) shows the performance of Return Capping against two of the baselines. We have not included the CVaR-PG baseline here as it was unable to discover any successful policy. The Lunar Lander environment illustrates the issues present with standard CVaR-PG methods. The CVaR-PPO baseline converges to a very conservative policy that does land on the left but is subsequently unable to improve, which can likely be attributed to the *blindness to success* issue. However, using

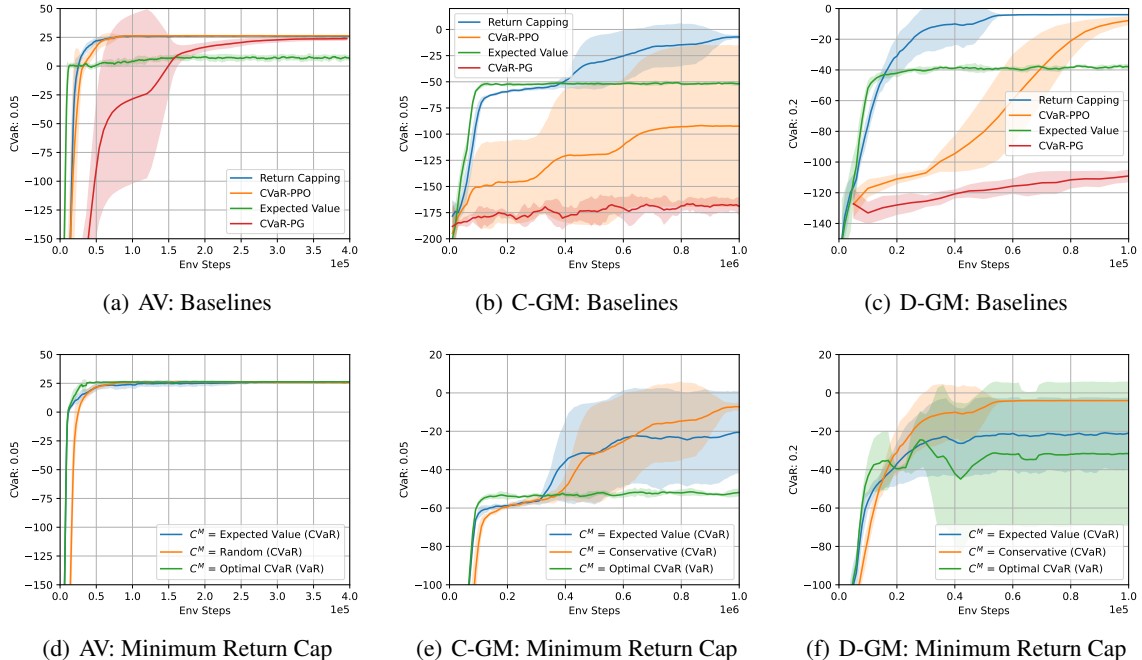

*Figure 5.* Autonomous Vehicle (AV) CVaR$_{\alpha=0.05}$, Continuous Guarded Maze (C-GM) CVaR$_{\alpha=0.05}$ and Discrete Guarded Maze (D-GM) CVaR$_{\alpha=0.2}$ performance. Comparing to Baselines and showing effect of different minimum return caps.

Return Capping, we are able to train a policy that both lands on the left-hand side but also is able to land without being overly conservative. Additionally, we have included the MIX baseline from (Luo et al., 2024). Given we require the risk-neutral policy to set $C^M$, we included the MIX baseline mixing this policy with a learnt risk-averse policy. However, we see from this baseline that this simply converges to the risk-neutral policy. This environment highlights one of the issues with the MIX baseline, being that if the risk-neutral and risk-averse policies are in direct opposition in some regions, learning the risk-averse policy may be made more difficult by the interference of the risk-neutral policy, and increases the probability of converging to this local optimal.

## 7. Conclusion

This paper proposes a reformulation of the CVaR optimisation problem by capping the total return of trajectories used it training, rather than simply discarding them. We have shown empirically that this method shows significant improvement to baseline CVaR-PG methods in a number of environments, although there is still space for additional experimentation in a broader range of environments. As we have shown, some environments are sensitive to the minimum cap value, and further environment testing may help to better identify environment characteristics that lead to this.

Further work could be done examining how Return Capping can be combined with other CVaR optimisation methods

such as those proposed in (Greenberg et al., 2022; Lim & Malik, 2022; Luo et al., 2024). Although these combination algorithms fell outside the scope of this research, examining whether performance improvements can be made here may be an interesting area to explore for future work.

## Acknowledgements

This work was supported by the EPSRC Centre for Doctoral Training in Autonomous Intelligent Machines and Systems [EP/S024050/1]. Lacerda and Hawes have received EPSRC funding via the "From Sensing to Collaboration" programme grant [EP/V000748/1].

## Impact Statement

This paper presents work whose goal is to advance the field of Machine Learning. There are many potential societal consequences of our work, none which we feel must be specifically highlighted here.

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

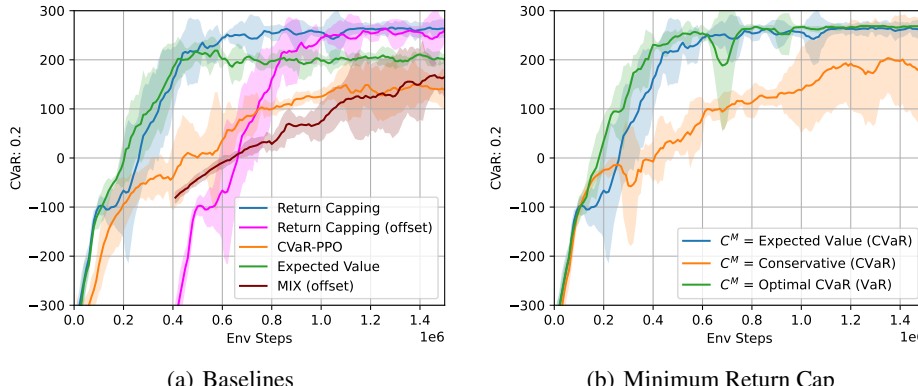

(a) Baselines        (b) Minimum Return Cap

*Figure 6.* Lunar Lander CVaR$_{\alpha=0.2}$ performance. Comparing to Baselines and showing effect of different minimum return caps. We have included MIX (Luo et al., 2024) as an additional baseline. The (offset) plots have been offset by the training steps required to learning the risk-neutral policy.

measures of risk. *Mathematical finance*, 9(3):203–228, 1999.

Bäuerle, N. and Ott, J. Markov decision processes with average-value-at-risk criteria. *Mathematical Methods of Operations Research*, 74:361–379, 2011.

Bellemare, M. G., Dabney, W., and Munos, R. A distributional perspective on reinforcement learning. In Precup, D. and Teh, Y. W. (eds.), *Proceedings of the 34th International Conference on Machine Learning*, volume 70 of *Proceedings of Machine Learning Research*, pp. 449–458. PMLR, 06–11 Aug 2017.

Brockman, G., Cheung, V., Pettersson, L., Schneider, J., Schulman, J., Tang, J., and Zaremba, W. Openai gym. *arXiv preprint arXiv:1606.01540*, 2016.

Chow, Y., Ghavamzadeh, M., Janson, L., and Pavone, M. Risk-constrained reinforcement learning with percentile risk criteria. *Journal of Machine Learning Research*, 18 (167):1–51, 2018.

Dabney, W., Ostrovski, G., Silver, D., and Munos, R. Implicit quantile networks for distributional reinforcement learning. In Dy, J. and Krause, A. (eds.), *Proceedings of the 35th International Conference on Machine Learning*, volume 80 of *Proceedings of Machine Learning Research*, pp. 1096–1105. PMLR, 10–15 Jul 2018. URL https://proceedings.mlr.press/v80/dabney18a.html.

Filar, J. A., Krass, D., and Ross, K. W. Percentile performance criteria for limiting average markov decision processes. *IEEE Transactions on Automatic Control*, 40 (1):2–10, 1995.

Greenberg, I., Chow, Y., Ghavamzadeh, M., and Mannor, S. Efficient risk-averse reinforcement learning. *Advances in Neural Information Processing Systems*, 35:32639–32652, 2022.

Hallak, A., Di Castro, D., and Mannor, S. Contextual markov decision processes. *arXiv preprint arXiv:1502.02259*, 2015.

Keramati, R., Dann, C., Tamkin, A., and Brunskill, E. Being optimistic to be conservative: Quickly learning a cvar policy. *Proceedings of the AAAI Conference on Artificial Intelligence*, 34(04):4436–4443, Apr. 2020. doi: 10.1609/aaai.v34i04.5870. URL https://ojs.aaai.org/index.php/AAAI/article/view/5870.

Kim, J.-H. and Min, S. Risk-sensitive policy optimization via predictive cvar policy gradient. In *Forty-first International Conference on Machine Learning*, 2024.

La, P. and Ghavamzadeh, M. Actor-critic algorithms for risk-sensitive mdps. *Advances in neural information processing systems*, 26, 2013.

Lim, S. H. and Malik, I. Distributional reinforcement learning for risk-sensitive policies. *Advances in Neural Information Processing Systems*, 35:30977–30989, 2022.

Luo, Y., Liu, G., Poupart, P., and Pan, Y. An alternative to variance: Gini deviation for risk-averse policy gradient. *Advances in Neural Information Processing Systems*, 36: 60922–60946, 2023.

Luo, Y., Pan, Y., Wang, H., Torr, P., and Poupart, P. A simple mixture policy parameterization for improving sample efficiency of cvar optimization. In *RLC*, pp. 573–592, 2024.

Ma, X., Xia, L., Zhou, Z., Yang, J., and Zhao, Q. Dsac: Distributional soft actor critic for risk-sensitive reinforcement learning. *arXiv preprint arXiv:2004.14547*, 2020.

Pan, X., Seita, D., Gao, Y., and Canny, J. Risk averse robust adversarial reinforcement learning. In *2019 International Conference on Robotics and Automation (ICRA)*, pp. 8522–8528. IEEE, 2019.

Pinto, L., Davidson, J., Sukthankar, R., and Gupta, A. Robust adversarial reinforcement learning. In Precup, D. and Teh, Y. W. (eds.), *Proceedings of the 34th International Conference on Machine Learning*, volume 70 of *Proceedings of Machine Learning Research*, pp. 2817–2826. PMLR, 06–11 Aug 2017.

Prashanth, L. and Ghavamzadeh, M. Variance-constrained actor-critic algorithms for discounted and average reward mdps. *Machine Learning*, 105:367–417, 2016.

Rigter, M., Lacerda, B., and Hawes, N. Risk-averse bayes-adaptive reinforcement learning. In Beygelzimer, A., Dauphin, Y., Liang, P., and Vaughan, J. W. (eds.), *Advances in Neural Information Processing Systems*, 2021.

Rockafellar, R. T., Uryasev, S., et al. Optimization of conditional value-at-risk. *Journal of risk*, 2:21–42, 2000.

Ruszczyński, A. Risk-averse dynamic programming for markov decision processes. *Mathematical programming*, 125:235–261, 2010.

Sato, M., Kimura, H., and Kobayashi, S. Td algorithm for the variance of return and mean-variance reinforcement learning. *Transactions of the Japanese Society for Artificial Intelligence*, 16(3):353–362, 2001.

Schneider, L., Frey, J., Miki, T., and Hutter, M. Learning risk-aware quadrupedal locomotion using distributional reinforcement learning. In *2024 IEEE International Conference on Robotics and Automation (ICRA)*, pp. 11451–11458, 2024. doi: 10.1109/ICRA57147.2024.10610137.

Schulman, J., Wolski, F., Dhariwal, P., Radford, A., and Klimov, O. Proximal policy optimization algorithms, 2017.

Sobel, M. J. The variance of discounted markov decision processes. *Journal of Applied Probability*, 19(4):794–802, 1982.

Stooke, A., Achiam, J., and Abbeel, P. Responsive safety in reinforcement learning by pid lagrangian methods. In *International Conference on Machine Learning*, pp. 9133–9143. PMLR, 2020.

Sutton, R. S. and Barto, A. G. *Reinforcement Learning: An Introduction*. The MIT Press, Cambridge, MA, 1998.

Tamar, A., Glassner, Y., and Mannor, S. Optimizing the cvar via sampling. In *Proceedings of the AAAI Conference on Artificial Intelligence*, volume 29, 2015.

Tang, Y. C., Zhang, J., and Salakhutdinov, R. Worst cases policy gradients. *arXiv preprint arXiv:1911.03618*, 2019.

Towers, M., Kwiatkowski, A., Terry, J., Balis, J. U., De Cola, G., Deleu, T., Goulão, M., Kallinteris, A., Krimmel, M., KG, A., et al. Gymnasium: A standard interface for reinforcement learning environments. *arXiv preprint arXiv:2407.17032*, 2024.

Wang, S. S. A class of distortion operators for pricing financial and insurance risks. *Journal of risk and insurance*, pp. 15–36, 2000.

Wu, L.-C., Zhang, Z., Haesaert, S., Ma, Z., and Sun, Z. Risk-aware reward shaping of reinforcement learning agents for autonomous driving. In *IECON 2023- 49th Annual Conference of the IEEE Industrial Electronics Society*, pp. 1–6, 2023.

Wu, Z., Tang, B., Lin, Q., Yu, C., Mao, S., Xie, Q., Wang, X., and Wang, D. Off-policy primal-dual safe reinforcement learning. *arXiv preprint arXiv:2401.14758*, 2024.

Yang, Q., Simão, T. D., Tindemans, S. H., and Spaan, M. T. Wcsac: Worst-case soft actor critic for safety-constrained reinforcement learning. In *Proceedings of the AAAI Conference on Artificial Intelligence*, volume 35, pp. 10639–10646, 2021.

## A. CVaR PPO

Given the Standard PPO loss as defined by

$$r(\theta) = \frac{\pi_\theta(a|s)}{\pi_{\theta_{\text{old}}}(a|s)}, \tag{21}$$

$$L^{\text{CLIP}}(\theta) = \mathbb{E}_{\tau \sim \pi_\theta} \left[ \min(r(\theta)\hat{A}, \text{clip}(r(\theta), 1 - \epsilon, 1 + \epsilon)\hat{A}) \right] \tag{22}$$

and the standard CVaR objective

$$J_\alpha(\pi_\theta) = \mathbb{E}_{\tau \sim \pi_\theta}[R(\tau)|R(\tau) \leq \text{VaR}_\alpha(R(\tau)]. \tag{23}$$

We introduce the CVaR-PPO loss

$$L_\alpha^{\text{CLIP}}(\theta) = \mathbb{E}_{\tau \sim \pi_\theta} \left[ \min(r(\theta)\hat{A}, \text{clip}(r(\theta), 1 - \epsilon, 1 + \epsilon)\hat{A})|R(\tau) \leq \text{VaR}_\alpha(R(\tau)) \right]. \tag{24}$$

Effectively, only doing the PPO update using the worst $\alpha$ proportion of trajectories. Similarly, the value function update is also done only on this proportion of trajectories. We show that this baseline works better than the standard CVaR Policy Gradient in some environments.

*Table 1.* Betting Game Hyperparameters

| PARAMTER | EXPECTED VALUE PPO | RETURN CAPPING | CVAR-PPO | CVAR-PG |
|---|---|---|---|---|
| CVAR $\alpha$ | 1 | 0.2 | 0.2 | 0.2 |
| NUM UPDATES | 200 | | | |
| NUM ENV STEPS PER UPDATE | 5000 | | | |
| EPOCHS PER BATCH | 5 | | | |
| SUB BATCH SIZE | 50 | | | 1000 |
| $\gamma$ | 0.99 | | | |
| LR | 1E-3 | | | |
| **PPO** | | | | |
| CLIP $\epsilon$ | 0.2 | | | |
| GAE $\lambda$ | 0.95 | | | |
| ENTROPY $\epsilon$ | 1E-5 | | | |
| **RETURN CAPPING** | | | | |
| CAP $\eta$ | | 0.2 | | |
| OPTIMAL CVAR (VAR) MINIMUM CAP | | 16 | | |
| EXPECTED VALUE (CVAR) MINIMUM CAP | | -16 | | |
| CONSERVATIVE (CVAR) MINIMUM CAP | | 0 | | |

## B. Betting Game

**State Space:** The agent's state is its current number of tokens (does not need to be an integer number), and its current turn.

**Action Space:** The agent's actions are to bet $\{0\%, 12.5\%, 25\%, 37.5\%, 50\%, 62.5\%, 75\%, 87.5\% 100\%\}$ of its current tokens.

**Reward:** The agent receives reward after each turn based on the total number of tokens gained/lost.

**Environment Dynamics:** Episode terminates after 6 bets or losing all tokens

*Table 2.* Autonomous Vehicle Road Time Cost Distributions

| ROAD TYPE | SMALL COST | MEDIUM COST | LARGE COST |
|-----------|-----------|-------------|------------|
| LANE | 7 | 7 | 8 |
| STREET | 4 | 5 | 11 |
| MAIN ROAD | 2 | 4 | 13 |
| HIGHWAY | 1 | 2 | 18 |

*Table 3.* Autonomous Vehicle Hyperparameters

| PARAMTER | EXPECTED VALUE PPO | RETURN CAPPING | CVAR-PPO | CVAR-PG |
|----------|--------------------|----------------|----------|---------|
| CVAR $\alpha$ | 1 | 0.05 | 0.05 | 0.05 |
| NUM UPDATES | 400 | | 200 | 66 |
| NUM ENV STEPS PER UPDATE | 1000 | | 2000 | 6000 |
| EPOCHS PER BATCH | 1 | | | |
| SUB BATCH SIZE | 50 | | | 300 |
| $\gamma$ | 0.99 | | | |
| LR | 1E-3 | | | |
| **PPO** | | | | |
| CLIP $\epsilon$ | 0.2 | | | |
| GAE $\lambda$ | 0.95 | | | |
| ENTROPY$\epsilon$ | 1E-5 | | | |
| **RETURN CAPPING** | | | | |
| CAP $\eta$ | | 0.6 | | |
| OPTIMAL CVAR (VAR) MINIMUM CAP | | 27 | | |
| EXPECTED VALUE (CVAR) MINIMUM CAP | | 6 | | |
| RANDOM (CVAR) MINIMUM CAP | | -256 | | |

## C. Autonomous Vehicle

**State Space:** Discrete x and y position of agent

**Action Space:** Move $\{Up, Down, Left, Right\}$

**Reward:** Table 2 outlines the distribution of costs in the Autonomous Vehicle environment. When the agent traverses a road, the cost is sampled from the distribution outlined, with 0.4, 0.3, 0.3 probability of receiving the Small, Medium or Large cost respectively. Reward of 80 for reaching the goal.

**Environment Dynamics:** Max number of env steps is 32

Figure 7 illustrates the performance in the Autonomous Vehicle without the outliers removed. These plots show the 95% confidence interval. Due to the 2 outlier runs, the mean performance of CVaR-PPO is substantially worse than the other methods.

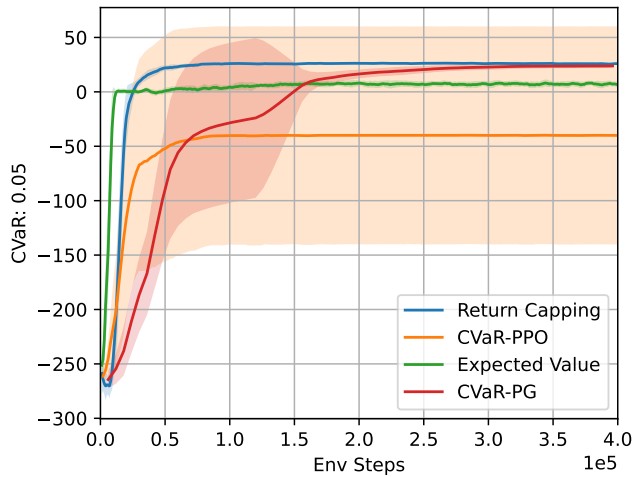

Figure 7. Autonomous Vehicle with outliers CVaR$_{\alpha=0.05}$ performance

Table 4. Guarded Maze (Continuous) Hyperparameters

| PARAMTER | EXPECTED VALUE PPO | RETURN CAPPING | CVAR-PPO | CVAR-PG |
|---|---|---|---|---|
| CVAR $\alpha$ | 1 | 0.05 | 0.05 | 0.05 |
| NUM UPDATES | 100 | | 200 | 66 |
| NUM ENV STEPS PER UPDATE | 10000 | | | |
| EPOCHS PER BATCH | 6 | | | |
| SUB BATCH SIZE | 50 | | | 500 |
| $\gamma$ | 0.99 | | | |
| LR | 1E-3 | | | |
| **PPO** | | | | |
| CLIP $\epsilon$ | 0.2 | | | |
| GAE $\lambda$ | 0.95 | | | |
| ENTROPY$\epsilon$ | 1E-5 | | | |
| **RETURN CAPPING** | | | | |
| CAP $\eta$ | | 0.2 | | |
| OPTIMAL CVAR (VAR) MINIMUM CAP | | -3.7 | | |
| EXPECTED VALUE (CVAR) MINIMUM CAP | | -54 | | |
| CONSERVATIVE (CVAR) MINIMUM CAP | | -151 | | |

# D. Guarded Maze (Continuous)

**State Space:** Continuous x and y position of agent

**Action Space:** Move $\{Up, Down, Left, Right\}$

**Reward:** -1 for each step in the environment. Reward of 16 for reaching the target zone. Guard is present with 20%. If the Guard is present, cost is sampled from an exponential distribution with a mean of 32.

**Environment Dynamics:** Max number of env steps is 161. The movement actions are all affected by added noise.

*Table 5.* Guarded Maze (Discrete) Hyperparameters

| PARAMTER | EXPECTED VALUE PPO | RETURN CAPPING | CVAR-PPO | CVAR-PG |
|---|---|---|---|---|
| CVAR $\alpha$ | 1 | 0.2 | 0.2 | 0.2 |
| NUM UPDATES | 100 | | 40 | 40 |
| NUM ENV STEPS PER UPDATE | 1000 | | 5000 | |
| EPOCHS PER BATCH | 6 | | | |
| SUB BATCH SIZE | 50 | | | 1000 |
| $\gamma$ | 0.99 | | | |
| LR | 1E-3 | | | |
| **PPO** | | | | |
| CLIP $\epsilon$ | 0.2 | | | |
| GAE $\lambda$ | 0.95 | | | |
| ENTROPY$\epsilon$ | 1E-5 | | | |
| **RETURN CAPPING** | | | | |
| CAP $\eta$ | | 0.2 | | |
| OPTIMAL CVAR (VAR) MINIMUM CAP | | -4 | | |
| EXPECTED VALUE (CVAR) MINIMUM CAP | | -40 | | |
| CONSERVATIVE (CVAR) MINIMUM CAP | | -90 | | |

# E. Guarded Maze (Discrete)

**State Space:** One-hot encoding of discrete state

**Action Space:** Move $\{Up, Down, Left, Right\}$

**Reward:** -1 for each step in the environment. Reward of 10 for reaching the target zone. Guard cost is sampled from a normal distribution $\mathcal{N}(0, 1) * 30$.

**Environment Dynamics:** Max number of env steps is 100.

*Table 6.* Lunar Lander Hyperparameters

| PARAMTER | EXPECTED VALUE PPO | RETURN CAPPING | CVAR-PPO | MIX |
|---|---|---|---|---|
| CVAR $\alpha$ | 1 | 0.2 | 0.2 | 0.2 |
| NUM UPDATES | 150 | | | |
| NUM ENV STEPS PER UPDATE | 10000 | | | |
| EPOCHS PER BATCH | 1 | | | |
| SUB BATCH SIZE | 50 | | | 2000 |
| $\gamma$ | 0.99 | | | |
| LR | 1E-3 | | | |
| **PPO** | | | | |
| CLIP $\epsilon$ | 0.2 | | | |
| GAE $\lambda$ | 0.95 | | | |
| ENTROPY$\epsilon$ | 1E-5 | | | |
| **RETURN CAPPING** | | | | |
| CAP $\eta$ | | 0.8 | | |
| OPTIMAL CVAR (VAR) MINIMUM CAP | | 270 | | |
| EXPECTED VALUE (CVAR) MINIMUM CAP | | 200 | | |
| CONSERVATIVE (CVAR) MINIMUM CAP | | -128 | | |

# F. Lunar Lander

**State Space, Action Space, Environment Dynamics:** As presented in OpenAI gym box2D (Brockman et al., 2016).

**Reward:** As presented in OpenAI gym box2D, but with additional $70 + \mathcal{N}(0, 1) * 100$ reward for landing on right size. The left and right regions are shown in Figure 8

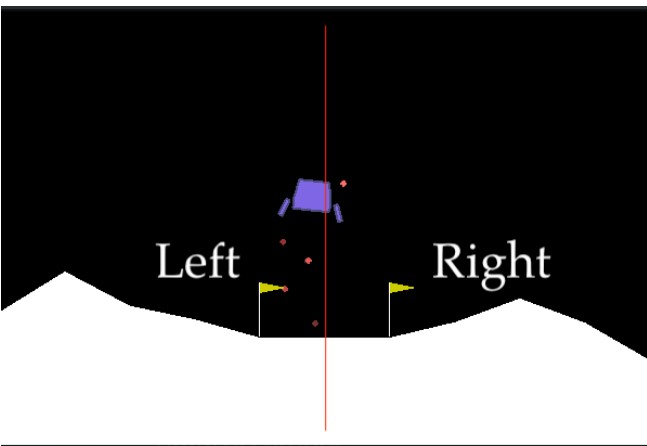

*Figure 8.* Lunar Lander environment from (Luo et al., 2023) showing divide between left and right landing zones

