# OpenReview forum: "Return Capping: Sample Efficient CVaR Policy Gradient Optimisation"
_ICML.cc/2025/Conference — ICML 2025 poster_

### Official Review · Reviewer_saRX · 2025-03-08

**Overall Recommendation:** 3

**Summary:**

This paper presents a new method for optimizing Conditional Value at Risk (CVaR) in reinforcement learning using policy gradients. Traditional CVaR policy gradient methods suffer from poor sample efficiency because they discard a large proportion of trajectories. The authors propose Return Capping, a novel approach that retains all trajectories but caps high returns at a threshold, ensuring better sample efficiency. Empirical results across multiple risk-sensitive environments show that Return Capping outperforms existing CVaR methods in both learning efficiency and robustness.

**Claims And Evidence:**

1. It is claimed that the solution of return capping objective results in the $CVaR_\alpha$ optimal policy if the cap is set correctly. I believe there are major technical flaws in the proofs of this; for example, the author claimed that (7) is the objective of return capping. However, in my perspective, if $C$ is independent of $\tau$, when $C<R(\tau)$ such that $C$ is chosen in the $min(\cdot)$ function, it will have no effect of optimizing $\pi_\theta$. Therefore, (7) is actually equivalent to (5), and the provided theoretical result is trivial, not being applied to the actual return capping algorithm authors proposed.

2. Using $VaR_\alpha(\pi_{\theta_{k-1}})$ at a starting of a training does seem to be a sensible choice.

**Essential References Not Discussed:**

I am not aware of essential references not discussed here.

**Experimental Designs Or Analyses:**

The experiments suggest superiority of the proposed method compared to baselines. However,
1. The experiments are only performed in relatively low-dimensional environments. Considering lack of theoretical contributions, I would expect more experiments and empirical analyses compared to previous works.
2. Different comparisons are conducted in different environments, e.g., offset, mix(offset) are only experimented in Lunar Lander. This lowers the statistical significance of the comparisons.

**Methods And Evaluation Criteria:**

Proposed method does make sense, but its not theoretically grounded. The algorithms are evaluated with CVaR, which is a trivial choice.

**Other Comments Or Suggestions:**

.

**Other Strengths And Weaknesses:**

- overall the paper is less dense compared to other top conference level papers. long, not important proof is included in the main paper, and environment describing figures, and result figures are organized in a less dense way.

**Questions For Authors:**

- What do you think about the issue above regarding the theoretical result?
- Is there any other way to formulate and justify the proposed algorithm?

**Relation To Broader Scientific Literature:**

This paper proposes a practical extension to previous CVar Policy optimization algorithms.

**Theoretical Claims:**

I have checked the correctness of the proof. The proof itself seems fine, although the result seems trivial and irrelevant to the proposed method.

---

> ### Author Rebuttal · Authors · 2025-04-01
>
> The main issue suggested in this review is flaws with the theoretical proof of the equivalence of Return Capping and standard CVaR PG optimisation. We would summarise the two points the reviewer raises as:
> - Given Eqn (7), when $C$ is selected in the $\min(\cdot)$, this trajectory will have no effect on the optimisation of $\pi_\theta$
> - Eqn (7) is not being applied to the actual return capping algorithm
>
> However, the above points are not the case, as we can show below.
>
> **Addressing the first point:**
>
> Given Equation 7
> $$J^C (\pi_\theta; C) = E_{\tau \sim \pi_\theta} [\min(R(\tau), C)].$$
>
> To compute policy gradients, we need to compute the gradient with respect to $\theta$
> $$\nabla_\theta J^C (\pi_\theta; C) = \nabla_\theta E_{\tau \sim \pi_\theta} [\min(R(\tau), C)].$$
> This can be expanded to the integral
> $$\nabla_\theta J^C (\pi_\theta; C) = \nabla_\theta \int_\tau P(\tau|\theta) \min(R(\tau), C).$$
> This integral highlights the flaw in the first point, as when either $R(\tau)$ or $C$ is selected in the $min(\cdot)$, the probability of that given trajectory is still dependent on $\theta$, and thus will have an effect on the policy optimisation. This addresses the first point made.
>
> **Addressing the second point:**
>
> From the equation above, we can move the gradient inside the integral and use a log-derivative trick to put the integral in the form
> $$\nabla_\theta J^C (\pi_\theta; C) = \int_\tau P(\tau|\theta) \nabla_\theta \log P(\tau|\theta) \min(R(\tau), C).$$
>
> We can then return this back to an expectation
> $$\nabla_\theta J^C (\pi_\theta; C) = E_{\tau \sim \pi_\theta} [\nabla_\theta \log P(\tau|\theta) \min(R(\tau), C)],$$
> and then we can reformulate the trajectory probability in terms of action probability
> $$\nabla_\theta J^C (\pi_\theta; C) = E_{\tau \sim \pi_\theta} [\sum_{t=0}^T \nabla_\theta \log \pi_\theta (a_t|s_t) \min(R(\tau), C)].$$
> From here, we can apply commonplace RL techniques such as using return-to-go, rather than using full episode returns, and using a Value function baseline.
> Note that return-to-go is computed as:
> $$\hat{R}^C_t =  \min(\sum_{t'=0}^T R(s_{t'}, a_{t'}, s_{t'+1}), C) - \min(\sum_{t'=0}^t R(s_{t'}, a_{t'}, s_{t'+1}), C).$$
> When incorporating both of these techniques, the gradient update becomes:
> $$\nabla_\theta J^C (\pi_\theta; C) = E_{\tau \sim \pi_\theta} [\sum_{t=0}^T \nabla_\theta \log \pi_\theta (a_t|s_t) (\hat{R}^C_t - V(s_t)].$$
> In practice, we used a Generalised Advantage Estimator to compute advantage. This gradient update can then be clipped, using the PPO algorithm, and then this is the exact gradient update used in Return Capping. As such the Return Capping gradient update is derived directly from Eqn (7), addressing the review’s second point.
>
> We are happy to provide any further clarification about the proof, or how the derivations above show that the reviewer’s main issues are unfounded.
>
> **Other Issues Raised**
>
> - In relation to the scale of the environments, we have discussed this thoroughly in our review to r3Le
> - The reason we have only included MIX [1] in Lunar Lander is that this baseline requires an optimal Expected Value policy. Lunar Lander was the only environment where Return Capping required the Expected Value policy to set $C^M$. However we could include this baseline in other environments - it generally just converged to the optimal Expected Value policy, similarly to in Lunar Lander. As explained in the paper, the Return Capping (offset) example was included to demonstrate the relative performance of Return Capping accounting for the environment steps required to train an optimal Expected Value policy.
>
> [1] Luo, Yudong, et al. "A simple mixture policy parameterization for improving sample efficiency of cvar optimization."

---

> > ### Comment · Reviewer_saRX · 2025-04-01
> >
> > Sorry for the incorrect review. I think I was confused about this; I am updating my score accordingly.

---

### Official Review · Reviewer_7ZeW · 2025-03-14

**Overall Recommendation:** 3

**Summary:**

The authors consider risk sensitive reinforcement learning where the goal is to optimize the $\alpha$-parameterized tail of the return distribution (on the lower end). Prior work proposed an approach known as conditional value at risk (CVaR) policy gradient, where the algorithm filters out all trajectories except the worst $\alpha$ fraction and does policy gradient on the remaining trajectories.

The authors note that this results in wasting a lot of samples, and propose an alternative equivalent formulation with a parameterized threshold for capping the return instead.  The authors show that for an appropriately chosen (only approximable) threshold for the return cap, the two approaches are equivalent. The paper evaluates the proposed approach against prior baselines in numerical experiments on toy domains and shows benefits to the new proposal.

**Claims And Evidence:**

Yes

**Essential References Not Discussed:**

n/a

**Experimental Designs Or Analyses:**

Yes, the authors consider an array of small scale toy environments to evaluate the new algorithm against baselines. The authors also modify the lunar lander environment to make it more interesting to see the difference between risk neutral versus risk sensitive optimization, which seems like a reasonable design.

**Methods And Evaluation Criteria:**

Yes

**Other Comments Or Suggestions:**

The relation between Eq (2) and (3) appears elementary but might be worth spelling out clearly as it may not be obvious when encountering for the first time.

**Other Strengths And Weaknesses:**

Strengths: Clear and concise algorithm proposed based on a novel but simple insight with numerical evidence.

Weaknesses: Most of the experiments are on very small toy domains.

**Questions For Authors:**

While the claim about sample efficiency sounds reasonable, it is not clear how that materializes in practice given the lack of sensitivity of the policy to the outcome beyond the threshold. Furthermore, the graphs appear to show a benefit for the final risk sensitive return, but not necessarily the speed of convergence (as one may expect for a drastic reduction in sample efficiency).

**Relation To Broader Scientific Literature:**

The authors show an intuitive and easy to implement alternative that is equivalent to prior work (filtering trajectories), but with better sample efficiency.

**Theoretical Claims:**

The main claim is in Proposition 4.1, which looks reasonable to me.

---

> ### Author Rebuttal · Authors · 2025-04-01
>
> In relation to the size of the environments presented, we have discussed this in our rebuttal to reviewer r3Le.
>
> **Sample Efficiency**
>
> For the specific question on sample efficiency, whilst there are improvements to sample efficiency using Return Capping compared to CVaR PG, it is unlikely to achieve improvements of magnitude $\frac{1}{\alpha}$ as by capping returns, we do lose some information about trajectories. However there is still information to be learnt from capped trajectories.
>
> Primarily, examples of trajectories that reach the cap are present in policy optimisation. In standard CVaR PG, we effectively only have negative examples of trajectories that resulted in low returns. Whilst we lose the relative performance between capped trajectories by capping returns, we still do get a gradient between the uncapped low performing trajectories and these capped trajectories. By training using capped trajectories, the policy is able to learn from positive examples (e.g. actions that result in the trajectory reaching the cap) rather than just having to rely on negative examples.
>
> In terms of speed of convergence observed in empirical results, in both Guarded Maze environments and the Lunar Lander environment, we see much faster convergence for Return Capping compared to the risk-sensitive baselines. The Expected Value policy does converge more quickly in all environments but this is unsurprising given it is learning from all uncapped trajectories. In the betting game, the CVaR PG policy does converge more quickly, albeit to a less optimal policy, than Return Capping. We suggest that this is likely due to the CVaR PG policy being a much more simplistic, conservative policy compared to the more optimal Return Capping policy. We agree that in the AV environment, there is minimal performance improvement over CVaR PPO*, but we suggest that the better performance of Return Capping in all other presented environments suggests it is a better method overall.
>
> *It should be noted that the plot in Figure 5a excludes CVaR PPO outliers that did not converge to the optimal CVaR policy (see Figure 7 in the Appendix for the unmodified plot), so although median performance is comparable to Return Capping, in this environment, CVaR PPO is less consistent at finding the CVaR optimal policy.

---

### Official Review · Reviewer_r3Le · 2025-03-16

**Overall Recommendation:** 3

**Summary:**

The authors address the problem of risk-sensitive policy optimization via policy gradient methods. They note several issues with the standard formulation of PG+CVaR which cause catastrophic losses in performance: specifically, because it discards the best trajectories by design, it is extremely difficult for an RL algorithm to encounter "lucky" transitions early in training from which it can learn a useful policy, resulting in extremely poor sample efficiency.

The paper proposes an alternative formulation for computing risk-sensitive policy gradients, by _capping_ the returns rather than subsampling the worst-case trajectories. They show that optimizing this capped-return formulation is equivalent to the standard formulation of CVaR under the condition that the cap is set at the VaR, and devise a practical algorithm to compute it by estimating the VaR online.

They show strong results with a practical implementation of this algorithm on top of PPO in risk-sensitive control baselines.

**Claims And Evidence:**

The authors claim that their method is better able to converge to a risk-sensitive policy. This does seem to be true in practice on the environments studied (where the CVaR of their method is substantially better than both the risk-neutral policy and prior work in risk-sensitive RL).

**Essential References Not Discussed:**

As mentioned previously I would recommend comparing against works that consider risk-sensitive control with off-policy approaches [1, 2].

[1] Yang, Qisong, et al. "WCSAC: Worst-case soft actor critic for safety-constrained reinforcement learning."
[2] Ma, Xiaoteng, et al. "Dsac: Distributional soft actor critic for risk-sensitive reinforcement learning."

**Experimental Designs Or Analyses:**

The experimental results are strong if somewhat limited; the most complex task studied is Lunar Lander which has some odd design choices in the experimental study:
> we modify the environment such that in addition to the zero-mean, high-variance random reward, landing on the right-hand side also results in an additional 70 reward.

I would suggest the authors find some more realistic or complex environments in which CVaR is helpful.

**Methods And Evaluation Criteria:**

The method is evaluated in several risk-sensitive RL settings, although a few (particularly Lunar Lander) are somewhat contrived in order to force a difference between risk-sensitive and risk-neutral policies. Evaluations are fair, although given the focus on sample complexity it also might have been nice to include an off-policy method (where CVaR-based RL algorithms have also been studied, see [1, 2]).

Generally, my main complaint with the evaluation is that the environments studied are very simple; considering more complex environments would greatly strengthen the paper's claims.

[1] Yang, Qisong, et al. "WCSAC: Worst-case soft actor critic for safety-constrained reinforcement learning."
[2] Ma, Xiaoteng, et al. "Dsac: Distributional soft actor critic for risk-sensitive reinforcement learning."

**Other Comments Or Suggestions:**

- The decision to exclude two outlier results from the AV results for the PPO baseline is odd; I would suggest using the original plot (and plot the median between seeds if there is a concern of outliers).
 - The results section writes out how many training runs of each baseline converged to which policies:
>For each method, we ran six seeds. None of CVaR-PG runs converged to a policy that reached the goal, and of the CVaR-PPO seeds, two converged to the CVaR-optimal policy, one converged to the risk-neutral policy, and three did not converge to goal-reaching policies

I would suggest conveying this information visually for all environments if it's important.

**Other Strengths And Weaknesses:**

- Learns a risk-neutral policy first (at least in Lunar Lander, see: (offset)), but it's not clear how this fits into the overall algorithm (maybe adding another algorithm block would be helpful for clarity).
 - The cap value is fixed over all trajectories, but it should probably be conditioned on initial state for any environments in which initial state has a large impact on returns (many practical environments).

**Questions For Authors:**

No additional questions; my main concerns are on the complexity of the environments studied.

**Relation To Broader Scientific Literature:**

The paper is well-positioned in the risk-sensitive RL setting, proposing a relatively simple theoretical modification to the risk-sensitive RL setting that induces a fairly large improvement in practice.

**Theoretical Claims:**

I looked over the proof for Proposition 4.1 and it seems reasonable, though I did not check it in detail.

---

> ### Author Rebuttal · Authors · 2025-04-01
>
> Thank you for your time spent reviewing this paper and we appreciate your feedback
>
> **Environment Complexity**
>
> The main issue raised in this review is the limited complexity of the environments used. Whilst we agree that more complex environments would benefit the paper, as far as we are aware, there is very limited work that has presented promising results for optimising static CVaR for episode return in more complex environments.
>
> Some related work does explore more complex environments. In Safe RL, such as [4], the main benchmarks are modified MuJoCo environments [7]. However, unlike in our work, in Safe RL, the objective is to maximise expected return subject to constraints on cost, where environments have a distinct cost and reward function. Some distributional RL work has been done in MuJoCo [5] and Atari [8]. However these works focus on dynamic CVaR, rather than static CVaR.
>
> The work we are aware of optimising for static CVaR referenced in the paper [1, 2, 6] as well as a similar work [3] suggested by Reviewer iMqv all focus primarily on comparatively similar scale environments to the examples presented in our paper. One of the exceptions are some modified MuJoCo environments in [1]. However, in practice, when optimising policies in these environments we found no distinction between optimal Expected Value and optimal CVaR policies. The other exception is the Atari game Asterix in [6]. This paper presents a modification to distributional DQN to optimise for static CVaR and shows promising results. However, results from [8] show that optimising for dynamic CVaR using distributional DQN in Asterix results in better Expected Value performance that a policy optimising for Expected Value, so it is ambiguous whether the improved CVaR performance in [6] was due to a distinct CVaR optimal policy being found, or due to this aforementioned result shown in [8].
>
> The main issue we have found in scaling to more complex environments is an issue inherent to return CVaR optimisation rather than specifically our method which is that for CVaR optimisation techniques to be relevant, the environment has to have distinct CVaR-optimal and Expected-Value-optimal policies. Generally, what we have observed is that one of these policies will be less complex to learn, and so in more complex environments, either the Expected Value optimal policy converges to the CVaR-optimal policy, or all CVaR optimisation methods converge to the Expected Value optimal policy.
>
> **Off Policy Methods**
>
> Thank you for pointing out the two papers [4, 5] as both are relevant to risk-sensitive RL and we will include them in reference to Related Work. However, as mentioned above, they are both optimising for different objectives compared to our work.
>
>
>
>
>
>
> **Other Strengths and Weaknesses**
>
> Addressing the two points the review raised in the strengths and weaknesses section:
> - The reason we have included (offset) In the Lunar Lander environment, is we found better performance by optimising for Expected Value and then using this policy to set $C^M$ as outlined in Section 4.1, even accounting for the additional environment steps required to train the optimal Expected Value policy (see Return Capping (offset)). In all other environments, we set $C^M$ based on a trivial policy that took no actions.
> - Conditioning the cap on the initial state is a sensible suggestion if the desired goal is to optimise for CVaR conditioned on the initial state.
>
>
>
>
>
> [1] Luo, Yudong, et al. "A simple mixture policy parameterization for improving sample efficiency of cvar optimization."
>
> [2] Greenberg, Ido, et al. "Efficient risk-averse reinforcement learning."
>
> [3] Kim & Min, “Risk-Sensitive Policy Optimization via Predictive CVaR Policy Gradient”
>
> [4] Yang, Qisong, et al. "WCSAC: Worst-case soft actor critic for safety-constrained reinforcement learning."
>
> [5] Ma, Xiaoteng, et al. "Dsac: Distributional soft actor critic for risk-sensitive reinforcement learning."
>
> [6] Lim, Shiau Hong, and Ilyas Malik. "Distributional reinforcement learning for risk-sensitive policies.”
>
> [7] Ji, Jiaming, et al. "Safety gymnasium: A unified safe reinforcement learning benchmark."
>
> [8] Dabney, Will, et al. "Implicit quantile networks for distributional reinforcement learning." International conference on machine learning.

---

### Official Review · Reviewer_iMqv · 2025-03-18

**Overall Recommendation:** 3

**Summary:**

This paper proposes a novel method for CVaR optimization in Reinforcement Learning. The proposed method caps trajectory returns by a certain value and maximize its expected value with respect to a policy. It is theoretically shown that, the maximizer of the proposed objective matches the conventional optimal CVaR policy if the capping threshold value $C$ is set to the VaR of the optimal CVaR policy. In practice, the capping threshold $C$ is approximated by a moving average of VaR of the learning policy. The proposed method requires to set the minimum capping threshold $C^M$ appropriately. The proposed method does not need to discard sample trajectories unlike naive baselines. The effectiveness of the proposed method is validated several numerical experiments.

## update after rebuttal
I keep my score from the following reasons.
- The authors answered adequately to my question that "$C^M$ seems more difficult to set in larger domains". However, I believe that ultimately the practical applicability meeds to be evaluated experimentally.
- Though the paper makes solid contribution, the absence of convergence analysis slightly hurts the quality of the paper.

**Claims And Evidence:**

In my view, the contributions of this paper are mainly twofold; (1) the proposal of the novel objective Eq. (7) and the establishment of the equivalence with the conventional objective (Proposition 4.1) and (2) the proposal of the practical method to optimize the proposed objective and its numerical validation. The first contribution is supported by the proof of Proposition 4.1 and the second contribution is supported in Section 5. The proposed method consistently performs better than baselines.

**Essential References Not Discussed:**

The following paper [1] is not referred, which does not naively discard the sampled trajectories but assigns "weights" to them and optimize CVaR for RL.
[1] Kim & Min, Risk-Sensitive Policy Optimization via Predictive CVaR Policy Gradient, ICML, https://proceedings.mlr.press/v235/kim24x.html.

**Experimental Designs Or Analyses:**

The experimental design seems to appropriate to validate the aforementioned two claims. However, since the simplicity of the environments, it is not fully convincing that the proposed method scales to the larger environments, where the $C^M$ is more difficult to set appropriately, i.e., Expected Value (CVaR) and Optimal CVaR (VaR). In addition, comparison with stronger baselines (e.g. [Greenberg et al., 2022] and [Luo et al., 2024]) is absent.

**Methods And Evaluation Criteria:**

The proposed method seems sound and the evaluation criteria makes sense to validate the aforementioned two claims, though the environments are relatively small and simple, and baselines are limited to Expected PG, CVaR-PG and CVaR-PPO.

**Other Comments Or Suggestions:**

N/A

**Other Strengths And Weaknesses:**

- Weakness
The absence of the convergence analysis of Algorithm 1.

**Questions For Authors:**

Do you think that the proposed method works well in the larger environments, such as Atari, where the $C^M$ seems more difficult to set appropriately? It would be grateful if you could discuss with some evidence.

**Relation To Broader Scientific Literature:**

The aforementioned contribution (1) is an interesting reformulation of CVaR optimization in RL. Broad class of RL methods could be applied to this formulation.

**Theoretical Claims:**

I checked the proof of Proposition 4.1 and it seems correct.

---

> ### Author Rebuttal · Authors · 2025-04-01
>
> Thank you for referencing [3], we will include it in Related Work
>
> **Setting Cap Minimum**
>
> Addressing the question on setting $C^M$ in larger environments, it is very possible to set a suitable $C^M$, irrespective of the complexity of the environment.
>
> We show in Proposition 1 that the Return Capping optimisation objective is equivalent to optimising for CVaR if $C$ is set to $VaR_\alpha(\pi^*)$, so we need to ensure that $C^M$, is less than or the equal to $VaR_\alpha(\pi^*)$.
>
> Given Eqn (17)
> $$CVaR_\alpha(\pi) \leq VaR_\alpha(\pi), \quad \forall \pi$$
> and Eqn (18)
> $$CVaR_\alpha(\pi) \leq CVaR_\alpha(\pi^*), \quad \forall \pi$$
> we know that the $CVaR_\alpha$ of any policy is necessarily less than the $VaR_\alpha(\pi^*)$. So we know that if we set $C^M$ to  $CVaR_\alpha$ of any policy, it will be less than $VaR_\alpha(\pi^*)$. This means we can take any policy and sample a set of trajectories and use this sampled $CVaR_\alpha$ to set $C^M$. This could be a random policy, or it could be a policy that maximises Expected Value.
>
> Even if an environment was sufficiently complex that it was not immediately apparent what an appropriate value of $C^M$ was, it would always be possible to use either the  $\text{CVaR}_\alpha$ of a random policy, or a policy trained in any manner to set $C^M$.
>
> Requiring a previously trained policy to set $C^M$ does increase the samples required for training. However, for all but the AV environment, Return Capping would outperform all baselines even accounting for the additional training required to learn an Expected Value optimal policy (shown as Return Capping (offset) in Lunar Lander as this was the only environment that required the cap to be set according to the Expected Value policy, but we could include this in all other environments as well).
>
> **Environment Simplicity**
>
> We have discussed issues with more complex environments further in our rebuttal to reviewer r3Le, but these issues arise more from challenges with optimising for CVaR in general, rather than any specific characteristics of Return Capping.
>
> **Additional Baselines**
>
> Addressing the lack of inclusion of [1, 2] as baselines, the reason we have not included [2] is because it requires the environment to be formulated as a Context-MDP where the context encapsulates the environment randomness. This limits the applications of the baseline as not all problems can be formulated as such. The reason we have only included [1] in Lunar Lander is that this baseline requires an optimal Expected Value policy. Lunar Lander was the only environment where Return Capping required the Expected Value policy to set $C^M$. However we could include this baseline in other environments - it generally just converged to the optimal Expected Value policy, similarly to in Lunar Lander.
>
> [1] Luo, Yudong, et al. "A simple mixture policy parameterization for improving sample efficiency of cvar optimization."
>
> [2] Greenberg, Ido, et al. "Efficient risk-averse reinforcement learning."
>
> [3] Kim & Min, “Risk-Sensitive Policy Optimization via Predictive CVaR Policy Gradient”

---

### Decision · Program_Chairs · 2025-05-01

**Decision:**

Accept (poster)

**Comment:**

This paper presents a clear and meaningful contribution to sample-efficient CVaR optimization in reinforcement learning. The authors introduce a novel approach that caps returns based on an estimated VaR, in contrast to conventional methods that discard high returns. This modification is shown to improve sample complexity. The method is partially supported both theoretically—demonstrating that, under an accurate VaR estimate, the optimal policy under the proposed objective aligns with the CVaR-optimal policy—and empirically in relatively simple domains.

A notable weakness lies in the treatment of the parameter $C^M$. While the paper offers some guidelines for selecting $C^M$, its theoretical justification remains limited, and the method’s performance is sensitive to this choice. Moreover, the appropriate value (strategy to set it) appears to vary significantly across domains (e.g., Figure 5(f) vs. Figure 6(b)). Nonetheless, the paper introduces a novel idea that is intuitively appealing and opens promising directions for future research in CVaR optimization.